# OpenReview forum: "When Agents Go Rogue: Activation-Based Detection of Malicious Behaviors in Multi-Agent Systems"
_ICML.cc/2026/Conference — ICML 2026 regular_

### Official Review · Reviewer_yCAv · 2026-03-08

**Soundness:** 3
**Presentation:** 3
**Significance:** 3
**Originality:** 3
**Overall Recommendation:** 4
**Confidence:** 5

**Summary:**

This paper proposes AcMAS, a method for security detection and mitigation in multi-agent systems (MAS) under attacks such as prompt injection, malicious tool use, and memory poisoning. The key idea is to extract representations from the final-layer activations of the underlying LLM in each agent, construct a benign prototype $\mu_{\text{normal}}$, measure deviations using cosine distance and reverse-direction detection, and then correct detected abnormal activations by steering them back toward the benign direction. The experiments cover synchronous and asynchronous MAS settings, five attack scenarios, and multiple open-source LLM backbones, and report both detection metrics and defense effectiveness. The results show that AcMAS significantly outperforms baseline methods in asynchronous settings, while reducing ASR to nearly zero under a relatively high TCR.

**Compliance With Llm Reviewing Policy:**

Affirmed.

**Final Justification:**

This paper proposes AcMAS, an activation-based framework for detecting and mitigating attacks in multi-agent systems via a benign prototype in activation space and activation-level steering. The approach is technically sound and clearly presented, and the results across five attack scenarios in both synchronous/asynchronous settings show strong detection and improved defense effectiveness while preserving task completion. The rebuttal addressed my main concerns by clarifying hook-based activation injection, adding baselines, and providing overhead/latency and benign-trace sensitivity analyses. Remaining limitations include lack of evaluation against adaptive attackers (activation mimicry) and reliance on white-box activation access/intervention. I therefore maintain my overall recommendation as Weak Accept (4/6) with higher confidence.

**Key Questions For Authors:**

Q1. How is the corrected activation injected into the final layer in the actual implementation? Does this require modifications to the inference engine or the model forward pass? Is it generally compatible with common open-source inference stacks?

Q2. If the attacker explicitly optimizes for activations that remain close to $\mu_{\text{normal}}$ (or even exploits the correction mechanism itself), would AcMAS’s detection and correction still be effective? Could the authors provide a simple adaptive attack experiment or at least a discussion?

Q3. What is the average latency introduced by extracting the final-layer activation at each step, computing the distance, and potentially regenerating the response? Can the method still operate in real time with 80 agents or at even larger scales?

Q4. How many benign traces are needed to construct $\mu_{\text{normal}}$? Does it need to be recalibrated across tasks or across different topologies? How sensitive is the method to the amount of benign data?

**Limitations:**

The paper’s applicability relies on strong white-box assumptions: access to internal activations and the ability to intervene at the final layer during generation for activation-level correction, which may not hold for closed-source LLM APIs or many production MAS stacks. The runtime overhead and latency of per-step activation extraction, detection, and potential regeneration/correction are not quantified, leaving real-time feasibility unclear at larger agent scales. Robustness to adaptive attackers (e.g., activation mimicry that minimizes distance to the normal prototype, or attacks exploiting the correction mechanism) is not evaluated and should be discussed. The “task completion rate” metric does not capture answer correctness; reporting task-specific accuracy (e.g., GSM8K accuracy, HotPotQA EM/F1) under defense would clarify potential trade-offs. Finally, releasing details of attack construction and defense mechanisms may enable stronger attacks; the authors should consider responsible disclosure (e.g., red-teaming notes, limits on releasing attack prompts/data, and safety guidelines for deployment).

**Strengths And Weaknesses:**

Strengths:

1、The method uses a benign prototype in activation space together with deviation-based detection, which is conceptually clear, formally well-defined, and simple and efficient when combined with threshold-based decision rules. The introduction of the momentum term $\alpha$ to balance adaptability and robustness is also practically appealing.

2、In the synchronous setting, the prototype is updated with momentum; in the asynchronous setting, a fixed prototype is used to avoid assumptions of temporal alignment. Overall, the design is well matched to the execution paradigms of the two settings.

3、Table 2 shows that isolation-based methods significantly reduce the task completion rate, whereas AcMAS’s activation correction reduces ASR while maintaining a high TCR, which better aligns with the practical collaborative requirements of MAS.

4、The evaluation covers five attack scenarios, both synchronous and asynchronous execution, multiple open-source backbones, and different MAS scales, and also includes sensitivity and ablation studies.

Weaknesses:

1、The current attacks mainly manifest as causing activations to deviate from the benign prototype. If an attacker explicitly optimizes the attack to minimize the distance to $\mu_{\text{normal}}$, the detection mechanism may fail.

2、The current experiments mainly compare against G-Safeguard and BlindGuard. It would be better to include more baseline methods, otherwise outperforming only two baselines may provide insufficient evidence.

3、The paper emphasizes real-time detection and mitigation, but I did not see an analysis of the latency and computational overhead introduced by activation extraction, distance computation, and regeneration/correction. This affects both the practicality of deployment and the fairness of comparison.

---

> ### Author Rebuttal · Authors · 2026-03-30
>
> We thank the reviewers for recognizing our design, the correction mechanism, and the evaluation. We group the comments as: **[W1 Q2 L3 Activation Mimicry], [W2 Baselines], [W3 Q3 L2 Overhead], [Q1 Activation Injection], [Q4 Benign Traces], [L1 White-box Assumption], [L4 Task Performance], and [L5 Responsible Disclosure].**
>
> ---
>
> ### **W1/Q2/L3 - Activation Mimicry Attack**
>
> Activation mimicry is a valid threat where attackers optimize inputs to keep malicious activations close to $\mu_{\text{normal}}$. While we acknowledge it as a limitation of AcMAS, as noted by NIST [1], such attacks however, adopt strong white-box assumptions of the access to the activations as well as the inputs for optimization. This kind of attack is more adopted as worst-case analysis rather than typical deployments [1]. While robustness to adaptive attacks remains an open challenge, a promising mitigation is latent adversarial training (LAT) [2], which uses adversarial activation perturbations to improve robustness. Overall, we will discuss the activation mimickry attack as a limitation of AcMAS with possible mitigation measures.
>
> [1] Vassilev et al. Adversarial Machine Learning: A Taxonomy and Terminology. *NIST AI 100-2, 2025.* [2] Sheshadri et al. Latent Adversarial Training Improves Robustness to Persistent Harmful Behaviors in LLMs. *TMLR, 2025.*
>
> ### **W2 - More Baselines**
>
> As suggested, we add two graph-based anomaly detection baselines: TAM (NeurIPS '23) and PERM (ICDM '23), developed for social and citation networks and compared in BlindGuard. Extended Table 1 is at: https://anonymous.4open.science/r/Figure-1F0E/Experiment_Add_baseline.png
>
> AcMAS outperforms all baselines. TAM and PERM underperform as graph-structure signals fail to capture subtle reasoning deviations, and GNN methods degrade in asynchronous settings. AcMAS leverages activation signals, remaining robust in both synchronous and asynchronous MAS.
>
> ### **W3/Q3/L2 - Overhead Analysis**
>
> As suggested, we add a detailed latency analysis, breaking down the avg runtime cost (single GPU A100) into: (1) activation extraction, (2) distance computation, (3) steering in response regeneration.
>
> **Table C: Computational Overhead and Latency Analysis**
> | Operation | Latency (ms) | Overhead (% of LLM Inference) |
> |---|---|---|
> | Activation Extraction | 2.3 | 0.15% |
> | Distance Computation | <1 | <0.07% |
> | Steering (Regeneration) | 185.3 | 12.4% |
>
> **Overhead.** AcMAS incurs negligible detection overhead (activation extraction&distance computation), accounting for only 0.22% of LLM inference time, enabling real-time monitoring. When anomalies are detected, correction requires an additional inference pass with activation steering, incurring 185.3ms(12.4%) overhead. This cost is only triggered for compromised agents, making it selective in practice.
>
> **Scalability.** AcMAS detection is inherently parallel, with per-agent activation and distance computations keeping cost constant under parallel execution. Correction is similarly localized, without blocking non-dependent agents. We note that in the worst case, correction latency accumulates linearly with full sequential agent dependency. Handling such dependency-aware scheduling to minimize end-to-end latency in highly sequential MAS topologies is an important direction, which we consider as future work.
>
> ### **Q1 - Activation Injection**
>
> Corrected activation injection is implemented via PyTorch forward hooks on the final transformer layer, adding the steering offset directly to the layer's activation output during the forward pass. This requires no modification to model weights or inference engine, and is fully compatible with HuggingFace LLMs.
>
> ### **Q4 - Number of Benign Traces**
>
> We empirically set the default value to 100 benign traces. Following your suggestion, we supplement a sensitivity analysis varying this as a hyperparameter on all datasets at https://anonymous.4open.science/r/benign-plot-0B1B/plot_benign_traces-1.png
>
> The Figure shows that few samples suffice for reliable prototype construction, with AcMAS’s performance stabilizing at ~50 traces and minimal gains beyond. While recalibration may be needed under domain shifts, AcMAS remains efficient and topology-agnostic, unlike GNN-based methods that require retraining.
>
> ### **L1 - White-box LLM Assumption**
>
> We have added a thorough discussion of the white-box LLM assumption. We kindly refer the reviewer to our detailed response (Reviewer 3-mnwt-W2).
>
> ### **L4 - Task Performance**
>
> We have conducted additional experiments on task performance as suggested, where AcMAS significantly outperforms SOTA baselines. We kindly refer the reviewer to our detailed response (Reviewer 2-wSMm-W2).
>
> ### **L5 - Responsible Disclosure**
>
> We appreciate this important reminder and will apply all suggested measures to ensure responsible disclosure.

---

> > ### Author Rebuttal · Reviewer_yCAv · 2026-04-03
> >
> > Thank you for the detailed rebuttal and the additional analyses. Overall, the rebuttal addresses my main concerns, and I maintain my original recommendation.

---

> > > ### Author Response · Authors · 2026-04-05
> > >
> > > Thank you very much for taking the time to review our rebuttal. We are very glad that it addressed your concerns. We sincerely appreciate your thoughtful comments and will incorporate them into the final version.

---

### Official Review · Reviewer_mnwt · 2026-03-11

**Soundness:** 2
**Presentation:** 3
**Significance:** 3
**Originality:** 2
**Overall Recommendation:** 4
**Confidence:** 3

**Summary:**

This paper addresses the problem of stealthy compromise in MAS. The authors propose AcMAS, an activation-based security framework that monitors agents’ internal hidden states rather than relying solely on output inspection. The method constructs a “normal prototype” representation from benign traces and detects anomalous agents based on activation divergence. Detected agents are then mitigated via activation steering, aiming to restore safe behavior without retraining model weights. The overall framework operates at inference time and is designed to preserve task completion while enhancing robustness.

**Compliance With Llm Reviewing Policy:**

Affirmed.

**Final Justification:**

The author has basically solved the problem I raised.

**Key Questions For Authors:**

The construction of $\mu_{\text{normal}}$ is highly dependent on specific benign samples and their distribution, but the paper lacks analysis of the stability of $\mu_{\text{normal}}$ under different data sources and domain distributions. Therefore, the current fit may experience an increase in false positives or a degradation in detection performance when the task category or domain shifts.

The proposed detector/mitigator assumes access to per-agent internal activations, which is often unavailable in real-world MAS deployments (closed-source commercial LLMs, hosted APIs, heterogeneous platforms), thereby limiting practicality and general applicability beyond fully white-box settings.

The paper's experiments do not clearly explain from which communication link the attack is triggered and propagates, making it difficult to determine whether the method truly utilizes MAS's unique interaction structure or is essentially equivalent to a scenario transfer that modifies the activation of a single LLM.

**Limitations:**

yes

**Strengths And Weaknesses:**

### Strengths

The paper highlights the limitations of output-level safeguards in MAS and monitoring internal activation dynamics as a more sensitive signal of compromise.

The experiments cover diverse attack categories (prompt injection, tool manipulation, memory poisoning) and report both security and utility metrics, enabling a balanced assessment of robustness versus task performance.

### Weaknesses

The construction of $\mu_{\text{normal}}$ is highly dependent on specific benign samples and their distribution, but the paper lacks analysis of the stability of $\mu_{\text{normal}}$ under different data sources and domain distributions. Therefore, the current fit may experience an increase in false positives or a degradation in detection performance when the task category or domain shifts.

The proposed detector/mitigator assumes access to per-agent internal activations, which is often unavailable in real-world MAS deployments (closed-source commercial LLMs, hosted APIs, heterogeneous platforms), thereby limiting practicality and general applicability beyond fully white-box settings.

The paper's experiments do not clearly explain from which communication link the attack is triggered and propagates, making it difficult to determine whether the method truly utilizes MAS's unique interaction structure or is essentially equivalent to a scenario transfer that modifies the activation of a single LLM.

---

> ### Author Rebuttal · Authors · 2026-03-29
>
> We sincerely appreciate your recognition of the novelty of activation-level monitoring and the comprehensiveness of our evaluation across diverse attack types. Based on your questions and suggestions, we number and respond to the weaknesses as follows: **W1 (Domain Shift), W2 (Open-weight LLM Assumption), and W3 (MAS Propagation & Method).**
>
> ---
>
> ### **W1 - Domain Shift**
>
> We acknowledge that sensitivity to distribution shift is a known challenge in prototype-based anomaly detection [1], where mismatched training and deployment distributions may increase false positives. This limitation is not unique to our method.
>
> A practical mitigation is few-shot recalibration [2], adapting the prototype with a small set of benign samples from the new domain. Longer-term, domain-adaptive prototype learning via meta-learning [3] offers another promising direction.
>
> Our additional analysis (Please see Reviewer 4 yCAv Q4) shows that AcMAS can achieve strong performance with a small number of benign traces (F1=0.91, AUROC=94.87), enabling few-shot recalibration for AcMAS in practice. We will include this as a limitation and discussion in the revised manuscript.
>
> **References:**
> [1] Cao et al. Anomaly Detection Under Distribution Shift. *ICCV 2023.*
>
> [2] Carvalho et al. Invariant Anomaly Detection under Distribution Shifts: A Causal Perspective. *NeurIPS 2023.*
>
> [3] Finn et al. Model-Agnostic Meta-Learning for Fast Adaptation of Deep Networks. *ICML 2017.*
>
> ---
>
> ### **W2 - Open Weight LLM Assumption**
>
> We agree that, in many real-world settings, access to internal activations may be limited. However, we clarify that the open weight LLM assumption aligns with a growing class of cost- and privacy-aware MAS deployments.
>
> **Cost-efficient heterogeneous MAS.** Recent work on LLM cascading [4] and routing [5,6] has established heterogeneous MAS — where simpler tasks go to local smaller models and complex tasks escalate to cloud APIs — as a standard cost-optimization pattern. In such architectures, local open-weight agents (e.g., LLaMA, Mistral) are fully accessible.
>
> **Data privacy.** Data-sensitive subtasks are typically routed to local on-premises models precisely because privacy regulations prohibit sending sensitive data to third-party APIs [7]. These local agents provide full activation access, making our method directly applicable to the most security-critical component of the MAS.
>
> **Activation-based safety: industry validation and the open-weight gap.** The value of activation-based safety research is further validated by the fact that leading providers such as Anthropic and OpenAI actively apply it to their own production models —identifying safety-relevant internal features including deception and dangerous content directly from model activations [8,9]. However, such provider-side defenses are internal and non-transferable: locally deployed open-weight models in heterogeneous MAS receive no such protection. Our work addresses precisely this gap, bringing activation-level defense to the open-weight agents that handle the most sensitive tasks in real-world pipelines.
>
> We will add the above discussion to a dedicated discussion and limitation section.
>
> [4] Chen et al. FrugalGPT. TMLR 2025.
>
> [5] Ong et al. RouteLLM. ICLR 2025.
>
> [6] Moslem et al. Dynamic Model Routing and Cascading for Efficient LLM Inference: A Survey. 2026.
>
> [7] Huang et al. A Middle Path for On-Premises LLM Deployment: Preserving Privacy Without Sacrificing Model Confidentiality. EMNLP 2025.
>
> [8] https://transformer-circuits.pub/2024/scaling-monosemanticity/ . Anthropic, 2024.
>
> [9] Gao, L., et al. Scaling and Evaluating Sparse Autoencoders (OpenAI). arXiv:2406.04093
>
> ---
>
> ### **W3 - MAS Propagation & Method**
>
> Following standard practice as G-Safeguard and BlindGuard, we model the MAS agent connectivity as a directed adjacency matrix, where each agent communicates only with its neighbors. Target agents are randomly selected from the agent pool. The attack is then instantiated according to the specific attack type and dataset — for prompt injection, malicious instructions are embedded in the agent's input; for tool manipulation, the tool-calling behavior is corrupted; for memory poisoning, the agent's memory is poisoned. The MAS then executes the task with multiple rounds of inter-agent communication. The attack effects (as outputs of compromised agents) propagate through the adjacency-defined communication links to downstream agents.
>
> This setup ensures attacks propagate through MAS communication rather than direct injection into each agent, demonstrating AcMAS’s effectiveness under real-world attack propagation. We will add this clarification to the experiment section.

---

> > ### Author Rebuttal · Reviewer_mnwt · 2026-04-01
> >
> > For W1, I hope to see more comprehensive experiments to confirm this.

---

> > > ### Author Response · Authors · 2026-04-02
> > >
> > > We really appreciate your rapid response and are glad to provide additional empirical analysis. Following your suggestion, we evaluated AcMAS under domain shift scenarios.
> > >
> > > **Experiment Setup.** We evaluate domain shift by treating each dataset as a distinct domain. We synthesize two domain-shift experiments, using CSQA as the source domain, and selecting InjecAgent and PoisonRAG as target domains (CSQA→InjecAgent, CSQA→PoisonRAG). These pairs correspond to the largest and second-largest centroid distances between the source and target domains. For each experiment, evaluation is performed on a fixed test set from the target domain.
> > >
> > > For centroid construction of AcMAS, we initialize $\mu_{\text{normal}}$ with 100 source-domain benign traces ($n_{\text{source}}$) and progressively add target-domain benign traces ($n_{\text{target}}$), updating $\mu_{\text{normal}}$ accordingly and re-evaluating AcMAS on the fixed test set. The ratio in each column header denotes $n_{\text{source}}:n_{\text{target}}$. We report F1 and FPR for each domain shift experiment in Tables E and F, respectively, and provide a **line chart** for clearer visualization of the performance trend https://anonymous.4open.science/r/domain_shift-0EBF/domain_shift.png
> > >
> > > Table E: F1 score and FPR under domain shift (CSQA → InjecAgent) with increasing numbers of target-domain benign traces ($n_{\text{target}}$).
> > >
> > > | Metrics |  |  | $n_{\text{target}} \; (n_{\text{source}} : n_{\text{target}})$ |  |  |  |  |
> > > |---------|--|--|--|--|--|--|--|
> > > |  | 0 (1:0) | 50 (1:0.5) | 100 (1:1) | 150 (1:1.5) | 200 (1:2) | 250 (1:2.5) | 300 (1:3) |
> > > | F1 ↑ | 0.49 | 0.49 | 0.49 | 0.58 | **0.86** | 0.89 | 0.92 |
> > > | FPR ↓ | 1.00 | 1.00 | 1.00 | 0.70 | **0.16** | 0.10 | 0.07 |
> > >
> > > Table F: F1 score and FPR under domain shift (CSQA → PoisonRAG) with increasing numbers of target-domain benign traces ($n_{\text{target}}$).
> > >
> > > | Metrics |  |  | $n_{\text{target}} \; (n_{\text{source}} : n_{\text{target}})$ |  |  |  |  |
> > > |---------|--|--|--|--|--|--|--|
> > > |  | 0 (1:0) | 50 (1:0.5) | 100 (1:1) | 150 (1:1.5) | 200 (1:2) | 250 (1:2.5) | 300 (1:3) |
> > > | F1 ↑ | 0.49 | 0.56 | **0.83** | 0.90 | 0.92 | 0.93 | 0.93 |
> > > | FPR ↓ | 1.00 | 0.83 | **0.19** | 0.08 | 0.07 | 0.06 | 0.06 |
> > >
> > > **Result Analysis.** As expected, AcMAS's performance degrades at the beginning of the domain shift (e.g. the F1 score in Tables are all only 0.49 when $n_{\text{source}}: n_{\text{target}}$ = 1: 0), as AcMAS is not explicitly designed for domain generalization and relies on learned representations that can be affected by distribution mismatch. However, AcMAS recovers fast during the progressive domain shift, i.e., when more target-domain benign traces are incorporated. As shown in the Tables, performance improves as the proportion of target-domain traces increases, with a sharp jump once target-domain traces outnumber source-domain traces (CSQA→InjectAgent 1:2, CSQA→PoisonRAG 1:1). Notably, the amount of target-domain data required for recovery correlates with the extent of the domain shift: *larger distance* (CSQA→InjecAgent) demands a higher proportion of target-domain traces (1:2).
> > >
> > > **Possible Adaptation Measures.** Importantly, the above results of AcMAS are achieved by default prototype updates, without any domain shift-specific adaptation. In real-world cases, when domain shift is known or detected, AcMAS can quickly adapt with low cost: by assigning higher weights to incoming samples in centroid computation, or directly discarding and recomputing the centroid, which requires only 50–100 traces. As shown here: https://anonymous.4open.science/r/benign-plot-0B1B/plot_benign_traces-1.png
> > >
> > > While we acknowledge that domain shift indeed has an impact on AcMAS initially, AcMAS can be naturally configured for lightweight and efficient adaptation. We will add the above analysis to the limitation and discussion section.

---

### Official Review · Reviewer_wSMm · 2026-03-13

**Soundness:** 2
**Presentation:** 2
**Significance:** 3
**Originality:** 3
**Overall Recommendation:** 4
**Confidence:** 3

**Summary:**

This work studies robustness in llm-based multi-agent systems. The authors build on prior works observing that neural activation patterns precede the emergence of harmful output behaviors, and propose a detection mechanism, AcMAS, which identifies attacks as distributional deviations. Rather than isolating the compromised agents, authors restore them by steering their activations toward a “normal” distribution gathered from the activations of agents behaving benignly. In their testing, the authors show the efficacy of the approach in detecting malicious agents, as well as a decrease of attack success rate (ASR) and an increase of task completion rate (TCR).

**Compliance With Llm Reviewing Policy:**

Affirmed.

**Final Justification:**

The author provide additional explanation specially on difference between their work and previous work.

**Key Questions For Authors:**

Please address the weaknesses.

**Limitations:**

yes

**Strengths And Weaknesses:**

**Strengths:**

1. I like the idea of restoring the compromised agents via activation steering rather than isolating them, allowing their computation to solve complex tasks.
2. The evaluation is performed across diverse open weights models (gpt-oss-20B, DeepSeek-V3, LLaMA3-8B, and Qwen3-30B-A3B).
3. The paper is well-structured and clearly written.

**Weaknesses:**

The paper lacks a few key points on the related works that better position the contribution of the work, and leaves some gaps in the evaluation.

1. [1] already observed the efficacy of detecting attacks through the activations of LLMs. I believe authors should better position their work compared to this.
2. While detecting the compromised agents is fundamental for a defense, final task performances are as well. Reporting only the TCR and ASR, without reporting the final performance on the task after the defense is applied, makes it unclear if the defense is actually useful in practical scenarios. Reporting the final performance before and after the defense provides a more comprehensive evaluation of the approach.
3. The paper is also missing a discussion on the limitations of the proposed defense. For example, the adaptive update of the distribution (eq. 13) can be leveraged by the attacker by systematically shifting their activations to poison the prototype and evade detection.

**Typos:**

1. In line 205, $\mathcal{H}^{(t)} = \{ h^{(t)}_i \}$ might need $i$ to be defined.

[1] Get my drift? Catching LLM Task Drift with Activation Deltas, Abdelnabi et al.

---

> ### Author Rebuttal · Authors · 2026-03-29
>
> We sincerely thank the reviewer for the insightful feedback. We appreciate the recognition of our core idea—restoring compromised agents via activation steering. We also acknowledge the concern regarding the positioning of our work and the evaluation completeness. In response, we have carefully revisited both aspects and provide clarifications and additional analysis below.
>
> For clarity, we organize the discussion into three parts: **W1 (Related Work), W2 (Task Performance), and W3 (Limitations and Adaptive Attacker)**, each corresponding to the key concerns raised.
>
> ---
>
> ### **W1 - Related Work**
>
> We thank the reviewer for pointing this out. We acknowledge that [1] (and our cited paper RevPRAG) share the same intuition of leveraging activation signals for attack detection.
>
> Our work differs in two key aspects. First, [1] focuses on task drift in single-LLM settings, whereas AcMAS targets a broader range of attacks in MAS, including prompt injection, tool misuse, and memory poisoning. Second, MAS introduces a unique attack surface—inter-agent communication—where compromised behavior can propagate across agents via compromised outputs, without direct malicious input. AcMAS is designed to detect such cross-agent propagation, which is beyond the scope of [1].
>
> We appreciate this feedback and will add this discussion and relevant citation to the revised manuscript.
>
> [1] Abdelnabi et al. Get my drift? Catching LLM Task Drift with Activation Deltas. *SaTML'2025*
>
> ---
>
> ### **W2 - Task Performance**
>
> The datasets fall into two categories: (1) attack-oriented (InjecAgent, PoisonRAG), which measure attack success without canonical ground truth, where ASR is appropriate; and (2) task-oriented (CSQA, GSM8K, HotPotQA), which provide ground-truth answers and support task accuracy evaluation. We therefore add experiments on the latter and report standard metrics: Accuracy (CSQA, GSM8K) and EM/F1 (HotPotQA). For comparison, besides the existing baselines G-Safeguard and BlindGuard, we further include Clean(no attack) and Attack-only(no defense) as performance upper and lower bounds.
>
> **Table A: Task performance under different defense methods**
>
> | Method | Defense | CSQA (Acc.) | GSM8K (Acc.) | HotPotQA (EM) | HotPotQA (F1) |
> |---|---|---|---|---|---|
> | Clean (no-attack) | — | 98.9% | 82.8% | 62.1 | 76.3 |
> | Attack-only (no-defense) | No | 16.1% | 9.0% | 4.4 | 7.1 |
> | G-Safeguard | Isolation | 78.9% | 63.9% | 35.5 | 42.0 |
> | BlindGuard | Isolation | 73.5% | 58.1% | 33.2 | 41.7 |
> | **AcMAS (Ours)** | **Correction** | **98.6%** | **81.3%** | **61.9** | **75.2** |
>
> As shown in Table A, the comparison between Clean and Attack-only shows that attacks severely degrade task performance across all datasets and metrics, confirming that MAS is highly vulnerable without defense. While G-Safeguard and BlindGuard block attack propagation, their isolation strategies disrupt interdependent reasoning, leading to performance well below the Clean setting. **By correcting instead of blocking**, AcMAS significantly outperforms both baselines, achieving near-clean performance across all benchmarks. These results demonstrate that AcMAS is not only effective in controlled evaluations but also practically useful, as it preserves task performance while mitigating attacks in realistic multi-agent workflows.
>
> ---
>
> ### **W3 - Limitations and Adaptive Attacker**
>
> We thank the reviewer for this insightful observation. Adaptive attack via activation shift is a valid concern, and also a well-recognized open challenge in adversarial machine learning.
>
> However, successfully executing this attack is non-trivial in the MAS context. The attacker must continuously inject carefully crafted inputs over an extended period to gradually shift the prototype, while simultaneously avoiding per-query detection—a conflicting requirement that significantly raises the operational cost and complexity. Prior work on online anomaly detector poisoning confirms that effective distribution shift requires sustained, high-volume injection, and that dynamic systems react to poisoning in ways that increase attack detectability [2, 3].
>
> A natural mitigation is an anomaly-gated update rule: only samples classified as benign can contribute to prototype updates, preventing compromised activations from corrupting the reference distribution. We'll include a dedicated discussion of this limitation and mitigation direction in the limitation and discussion section.
>
> [2] Kravchik et al. Practical evaluation of poisoning attacks on online anomaly detectors in industrial control systems. *Computers & Security*, 2022.
>
> [3] Korycki and Krawczyk. Adversarial concept drift detection under poisoning attacks for robust data stream mining. *Machine Learning*, 2023.
>
> **Typo.** In line 205, $i$ in $\mathcal{H}^{(t)} = h_i^{(t)}$ indexes each agent; we will clarify this notation in the revision.

---

> > ### Author Rebuttal · Reviewer_wSMm · 2026-04-03
> >
> > Thanks for the rebuttal. I will raise my score.

---

> > > ### Author Response · Authors · 2026-04-05
> > >
> > > Thank you very much for considering our rebuttal and for raising the score. We sincerely appreciate your insightful feedback and will incorporate the rebuttal points into the final manuscript.

---

### Official Review · Reviewer_9jn9 · 2026-03-15

**Soundness:** 3
**Presentation:** 3
**Significance:** 3
**Originality:** 3
**Overall Recommendation:** 5
**Confidence:** 4

**Summary:**

Existing MAS security assumes semantically explicit attacks and explicit graph-based modeling of
the MAS topology and interactions. This work addresses the limitations of the existing approach
by proposing the AcMAS framework, which finds stealthy attacks by spotting deviations in activation
patterns regardless of synchronicity. AcMAS also restores the functionality by steering back to
the normal patterns, instead of isolation or removal of the malfunctioning agent, thus avoiding
disruptions.

Attacks on MAS can be agent-level, communication-level, or system-level, in both synchronous and
asynchronous executions. Compromised agents can propagate the error, resulting in a cascade of failures.
Current defenses typically consider overall dynamics and communication, ignoring the internal reasoning
of the agents. On the other hand, AcMAS uses the correlation of the activation patterns of compromised
agents to detect those agents. They characterize the normal activation, updating it as time progresses,
and then consider the distance of the observed activation from it, to mark the suspicious agent. This is
a topology-agnostic approach.

Empirical evaluation shows high effectiveness against stealthy attacks, especially in asynchronous
cases, and achieves the completion of more tasks than isolation. It can operate on various architectures.
The evaluation proceeds for 3 scenarios, for synchronous and asynchronous cases each. AcMAS outperforms
all the other tested methods in each checked setting. AcMAS is scalable and generalizable. They also
conducted an ablation study and sensitivity analysis, the latter still showing the superiority of AcMAS
over the other tested methods.

**Compliance With Llm Reviewing Policy:**

Affirmed.

**Key Questions For Authors:**

Can you suggest merging AcMAS with other approaches to further expand and improve the defense capabilities?

In what cases is AcMAS less applicable or less efficient than the classic approaches? When shall
an engineer employ this method?

**Limitations:**

Comments for authors

Contents wise:

Some technical background about features, queries, cosine (innner-product) distance, etc. would help.


Typos and writing:

Line 843 in Appendix C seems to be excessive.

**Strengths And Weaknesses:**

Strengths:

The paper proposes a novel and efficient attack detection and resolution method. The method is clearly
described and shows clear superiority in the experimental evaluation against classis methods.

This method can both inspire new methods, both novel ones and combinations of existing ones. The provided
method is both conceptually deep and practically promising.


Weaknesses:

The paper might benefit from presenting more intuition as to the proposed method and from discussing
its downsides explicitly.

---

> ### Author Rebuttal · Authors · 2026-03-29
>
> We sincerely thank the reviewer for the positive feedback. We appreciate your recognition of the novelty and efficiency of our method. For clarity, we organize the comments into W (weakness), Q (question), and L (limitation), and further group them as **W1 (intuition and limitations)**, **Q1&Q2 (Collaborative Defense and Deployment)** and **L1 (Technical background)**, and respond accordingly. We refer to the reviewers in order as Reviewer 1–4 and use this consistent W/Q/L labeling scheme for cross-referencing across reviews.
>
> ---
>
> ### **W1 - Intuition & Limitation**
>
> The core intuition of AcMAS is that attacks on agents manifest in their internal activation patterns before they surface in observable outputs. Just as a person's cognitive state changes before their behavior does, a compromised agent's reasoning process deviates from normal patterns at the activation level, even when its textual output appears semantically plausible. This motivates our design: rather than inspecting outputs or modeling the interaction graph, AcMAS monitors the antecedent internal activation patterns inside each agent. Normal agent reasoning forms a stable distribution in activation space, captured by μ_normal. Attacks — whether prompt injection, tool manipulation, or memory poisoning — shift an agent's activation away from this distribution, which AcMAS detects as a divergence signal (Eq. 11–12) and corrects via activation steering (Eq. 14–15), restoring normal reasoning before malicious behavior propagates through the system.
>
> We will add a dedicated section on Discussion and Limitations, including: (1) Collaborative Defense, and Deployment (Q1, Q2), (2) Adaptive and adversarial attacks (Reviewer 2 wSMm–W3, Reviewer 4 yCAv-W1), (3) Domain shift (Reviewer 2 wSMm –W1), and (4) Open-weight LLM assumption (Reviewer 3 mnwt –W2).
>
> ---
>
> ### **Q1&Q2 - Collaborative Defense and Deployment**
>
> While designed as a MAS-specific defense leveraging activation-level signals, AcMAS is complementary to general LLM defense methods that operate on alternative signals, including input filtering, output monitoring, and system-level control mechanisms. Text-level guardrails (e.g., prompt filtering and output monitoring [1,2]) are typically deployed at the level of individual LLMs or agents to detect direct attacks based on input/output signals, and are effective at capturing malicious content in local agents. In contrast, AcMAS enables cross-agent defense by capturing and mitigating attack propagation through analysis of internal activation states.
>
> These complementary properties also clarify deployment trade-offs. AcMAS achieves highly efficient detection (0.22% of inference time; see Reviewer 4 yCAv–W3 Table) by operating directly on activation signals without requiring additional LLM inference, unlike text-based defenses that often rely on auxiliary LLM calls. The primary trade-off lies in applicability: AcMAS requires access to internal activations and is thus best suited for self-hosted or open-weight models, whereas text-based methods are more readily applicable in black-box or API-only settings. We provide further discussion on this assumption in (Reviewer 3 mnwt–W2).
>
>
> [1] Wang et al. Defending against prompt injection with DataFilter. *arXiv:2510.19207*, 2025.
>
> [2] Zheng et al. Judging LLM-as-a-Judge with MT-Bench and Chatbot Arena. *NeurIPS 2023*.
>
> ---
>
> ### **L1 - More Technical Background**
>
> We thank the reviewer and will add the following clarifications to **Section 3.2 and 3.3**:
>
> - **Query construction:** Each agent's input $x_i^{(t)}$ concatenates the task prompt, system prompt, and historical outputs from neighboring agents.
> - **Activation feature:** The hidden state of the last token at the final transformer layer is extracted as a $d$-dimensional vector $h_i^{(t)} \in \mathbb{R}^d$ (Eq. 6); across all $N$ agents these form an activation matrix $H^{(t)} \in \mathbb{R}^{N \times d}$ (Eq. 7).
> - **Cosine distance:** Eq.11 quantifies angular deviation from the normal prototype. Small $\delta$ indicates normal reasoning, while large $\delta$ suggests anomalous or compromised behavior.
>
> - **Outlier detection:** An agent is flagged if $\delta_i^{(t)} > \tau_{\text{detect}}$ (Eq. 12), where the threshold $\tau_{\text{detect}} = 0.12$ is calibrated by maximizing F1 on a validation set (Appendix D, Table 4).
>
> **Typo.** Thank you for catching this. We will remove the redundant phrase "asynchronous detection" and have carefully proofread the manuscript to prevent similar issues.

---

> > ### Author Rebuttal · Reviewer_9jn9 · 2026-04-04
> >
> > I would suggest adding the fragments of your rebuttal to the paper should the paper be accepted.

---

> > > ### Author Response · Authors · 2026-04-05
> > >
> > > Thank you very much for your positive feedback. We sincerely appreciate your constructive suggestions and will carefully incorporate the rebuttal content into the final version.

---

### Decision · Program_Chairs · 2026-04-30

**Decision:**

Accept (regular)

**Comment:**

The reviewers all basically agree on the strengths of the paper, finding the method introduced to be novel and potentially generative, the paper as a whole to be technically sound, and the evaluation strategy to be sufficiently broad and comprehensive to justify the paper's conclusions. The reviewers do raise a number of objections that the authors deal with comprehensively in the rebuttals, and each of the reviewers emphasises that the authors have basically solved the problems that they've raised. With all this in view, this seems to be a clear accept.